# Associations between Lifestyle Habits, Perceived Symptoms and Gastroesophageal Reflux Disease in Patients Seeking Health Check-Ups

**DOI:** 10.3390/ijerph18073808

**Published:** 2021-04-06

**Authors:** Chiu-Hua Chang, Tai-Hsiang Chen, Lan-Lung (Luke) Chiang, Chiao-Lin Hsu, Hsien-Chung Yu, Guang-Yuan Mar, Chen-Chung Ma

**Affiliations:** 1Nursing Department, Kaohsiung Veterans General Hospital, Kaohsiung 813414, Taiwan; cchchang@vghks.gov.tw; 2Department of Healthcare Administration, I-Shou University, Kaohsiung 82445, Taiwan; 3College of Management, Yuan Ze University, Taoyuan 32003, Taiwan; scts1215@yahoo.com.tw (T.-H.C.); lukech@saturn.yzu.edu.tw (L.-L.C.); 4Health Management Center, Kaohsiung Veterans General Hospital, Kaohsiung 813414, Taiwan; clhsu@vghks.gov.tw (C.-L.H.); hcyu@vghks.gov.tw (H.-C.Y.); 5Center for Geriatrics and Gerontology, Kaohsiung Veterans General Hospital, Kaohsiung 813414, Taiwan; 6Department of Nursing, Meiho University, Pingtung 912009, Taiwan; 7Department of Business Management, National Sun Yat-sen University, Kaohsiung 80424, Taiwan; 8Division of Gastroenterology and Hepatology, Department of Internal Medicine, Kaohsiung Veterans General Hospital, Kaohsiung 813414, Taiwan; 9Graduate Institute of Health Care, Meiho University, Pingtung 912009, Taiwan; philipmar0119@gmail.com; 10Kaohsiung Municipal United Hospital, Kaohsiung 80457, Taiwan

**Keywords:** gastroesophageal reflux disease, lifestyle habits, perceived symptoms

## Abstract

Gastroesophageal reflux disease (GERD) is one of the most common diseases. It mainly causes the stomach contents to flow back to the esophagus, thereby stimulating the esophagus and causing discomfort. From the results of our research, we intend to provide the general public with information related to preventing gastroesophageal reflux disease and medical personnel with information on the treatment and care of patients with gastroesophageal reflux disease. This study aimed to investigate the association of lifestyle habits and perceived symptoms on GERD in patients who underwent routine health check-ups. This study was conducted as a retrospective cross-sectional design to collect GERD cases from the medical records containing the health questionnaires and the report of endoscopic findings on the day of the health check-up. A total of 5653 patients were enrolled between 1 January 2016, and 31 December 2018. About 60.2% (*n* = 3404) of patients with GERD were diagnosed based on endoscopic findings. Descriptive and multivariate logistic regression analyses were performed to identify the risk factors of the development of GERD. The results of the multivariate logistic regression analysis showed that age, sex, waist circumference, Areca catechu chewing habit, sleep disorders, otolaryngology symptoms, and hepatobiliary and gastrointestinal symptoms were significantly associated with GERD. In this study, our results can be used as a reference for public health care and clinicians. Because most GERD cases can be controlled and prevented by lifestyle modifications, health professionals should always obtain a detailed history regarding symptoms and lifestyle habits associated with GERD.

## 1. Introduction

With the progress of socioeconomic standards, people’s lives move at a faster pace. With the increasing stress at work and changes in dietary habits, gastroesophageal reflux disease (GERD) has become the major disease of affluence. In recent years, countries worldwide have discovered in clinical research that the incidence rate of GERD has been increasing each year. Over the past 10 years, the Chinese population with GERD increased, but that with peptic ulcers has increased even more. GERD seems to have become the most common disease in the digestive system. The incidence rates of gastroesophageal reflux symptoms and disease also show increasing trends in Taiwan [1,2].

GERD is one of the most common diseases. It mainly causes the stomach contents to flow back to the esophagus, stimulating it and causing discomfort. Reflux esophagitis, which is due to corrosive damage caused by stomach acid, is already known to be the inducing agent of Barrett’s esophagus. The rate of development of esophageal cancer in patients with Barrett’s esophagus is 30–60 times higher than that in the general population [3,4]. In the past, the incidence of GERD in Western countries was approximately 2–4 times higher than that in Asian countries. This is generally believed to be caused by different eating habits and body shapes. However, in the past two decades, studies have found a significant increase in the incidence of GERD in Asian countries. This is not due to advances in screening techniques and data analysis, but rather to the increase in the prevalence of GERD, further leading to an increased prevalence of esophageal cancer [5,6]. Research has shown that the number of hospitalizations due to GERD in Taiwan tends to increase [7]. However, most past research studies were conducted in Western countries or in symptomatic patients. Few cases of GERD diagnosed using gastroscopy after health check-up have been reported. For the above-mentioned reasons, this study aimed to examine the influencing factors including lifestyle habits and perceived symptoms on GERD.

This study mainly collected cases of GERD confirmed with gastroscopy in health check-up. We analyzed the influencing factors of GERD by using a health information questionnaire and health check-up parameters. Finally, the association of demographics, lifestyle habits and perceived symptoms variables were analyzed. We also aimed to provide the public with information related to preventing GERD. Therefore, the specific objective of this study was to examine the association of lifestyle habits and perceived symptoms on GERD in patients who underwent endoscopy in their health check-ups.

## 2. Materials and Methods

### 2.1. Data Sources and Study Samples

This study used data from a health screening database of a medical center in southern Taiwan. The endoscopic diagnosis for esophagitis or GERD in this study was performed by four experienced endoscopists. All patients would know the results of endoscopic findings within the same day after their health check-up, as well as other results such as laboratory or images at the end of health check-up. The health check-ups were self-funded by patients. The patients with reflux esophagitis were classified as mild (grade A and B) or severe (grade C and D) according to the Los Angeles classification, while the rest were classified as normal. The design of this study was a retrospective cross-sectional study. This study only collected medical records from health check-up participants with complete medical records who have received gastroscopies. Their medical records containing the health questionnaire and the report of endoscopic findings were collected for analysis. The patients underwent gastroscopy in the health management center of the medical center between 1 January 2016, and 31 December 2018. Each patient would receive a health questionnaire by mail prior to the health check-up. They were required to complete the questionnaire and submit it for review before their health check-up. The main purpose of the questionnaire is to provide doctors with a quick confirmation of the health status of the patients, so the answers answered by the patients are not final. The last step is to proofread the questionnaire from the medical records to increase the accuracy. Then, their medical records were used to connect the information from the health questionnaires with the gastroscopic report data.

Furthermore, three stages of data matching were conducted. First, all the data was checked, and the incidence and severity of GERD found on gastroscopic examination were recorded in detail. In the health management center, all gastroscopy exams were performed by narrow band imaging (NBI) to improve GERD diagnosis accuracy. Second, three gastroenterologists from the gastroenterology department of the medical center were invited to compare and verify randomly sampled data according to the method suggested by Worster and Haines [8]. After verification, we found that the health information questionnaires of 150 cases were read incorrectly by the computer because the entries were too lightly written. The data were immediately compared, corrected, and logged. Finally, in order to improve the accuracy of the data, three physicians were invited again to jointly select 10% of the samples in the same way, and 565 samples were selected for the second verification. After verification, no data errors were found. The logged data in this study were 100% consistent. This study was approved by the institutional review board of the Kaohsiung Veterans General Hospital on 27 December 2018 (approval No. VGHKS19-CTI-06). The data were collected in the form of a review of the medical examination report, without any medical behavior or treatment for each case. Since this study was minimal risk and obtained from legitimate databases, the review board waived the need for consent.

### 2.2. Dependent Variables

Upper gastrointestinal endoscopy is the best diagnostic method for determining the presence and severity of esophagitis, and can be used to exclude other upper gastrointestinal causes. The Los Angeles classification of the severity of esophageal inflammation classified the cases into grades A to D, and multivariate logistic regression was used to classify the cases into normal, mild and severe. The classification criteria were as follows: grade A and B indicated mild degree, and grades C and D indicated severe degree, and the rest were classified as normal.

### 2.3. Independent Variables

The case data are divided into three parts as follows: demographic variables (sex, age, waist circumference, body mass index [BMI]), lifestyle habits variables (drinking, eating, smoking, *Areca catechu* (Palmaceae family, commonly known as betel nut or areca nut) chewing habits, and sleep disorders), and perceived symptoms variables (otolaryngological, cardiovascular, respiratory, renal urinary, and hepatobiliary and gastrointestinal symptoms). Darvishmoghadam et al. [9] conducted a cross-sectional study of GERD in Iran and found that occupation had no significant effect on GERD (*p* = 0.15). In addition, Alsulobi et al. [10] investigated the morbidity rate, main characteristics, and risk factors of GERD in 302 Arabian patients and found that occupational status had no significant influence on the occurrence of GERD (*p* > 0.05). As previous research showed that occupation had no correlation with the occurrence of GERD, occupation was excluded from the demographic variables in this study.

### 2.4. Statistical Analyses

This study used IBM SPSS Statistics version 20 (IBM Corp. Released 2011. IBM SPSS Statistics for Windows, Version 20.0. IBM Corp., Armonk, NY, USA) to process the data. The data analysis methods used were as follows: (1) Descriptive statistics: data were collected to describe the numerical distribution of all the variables in terms of frequency distribution, percentage, and so on. (2) Inferential statistics: on the basis of the nature of the variables in this study, we used multivariate logistic regression to examine the relationship between the independent variables (demographic, lifestyle habits and perceived symptoms variables) and the dependent variables to identify the risk factors of GERD severity.

## 3. Results

### 3.1. Basic Characteristics of the Study Samples

The patient characteristics are listed in Table 1. A total of 5653 patients were enrolled, of whom 39.8% were classified as normal, 53.1% had a mild degree of GERD, and 7.1% had severe GERD. In terms of sex, most patients with GERD were male (65.9%), while female were less (34.1%). Age was categorized into three groups: aged ≦ 30 (3.4%), aged 31–60 (71.6%), and aged > 60 (25.0%). Approximately 3.9% of subjects had a BMI < 18.5 kg/m^2^, followed by 18.5 ≤ BMI < 24 (44.8%), 24 ≤ BMI < 35 kg/m^2^ (50.7%) and a BMI ≥ 35 kg/m^2^ (0.6%). We defined underweight as a BMI < 18.5 kg/m^2^, normal as 18.5 ≤ BMI < 24 kg/m^2^, obesity as 24 ≤ BMI < 35 kg/m^2^, and morbid obesity as a BMI ≥ 35 kg/m^2.^ The waist circumference of most patients was <90 cm, accounting for 72.4%, followed by 27.6% of other patients with a waist circumference ≧90 cm.

### 3.2. Multivariate Logistic Regression Analysis of the Factors Related to GERD

Table 2 shows the results of the multivariate logistic regression analysis for the mild degree and severe degree as compared to normal patients (reference). Over all, age, sex, waist circumference, *Areca catechu* chewing habit, sleep disorders, otolaryngology symptoms, and hepatobiliary and gastrointestinal symptoms were significantly associated with GERD.

When all variables were adjusted, the patients between the ages of 31 to 60 were positively associated with the mild degree of GERD (adjusted odds ratio [AOR]: 1.132, 95% confidence interval [CI]: 0.540–0.993). Meanwhile, other age groups were not related with GERD. Compared with females, males were associated with a lower risk of the mild degree of GERD (AOR: 0.842, 95% CI: 0.732–0.969). Compared with the patients with waist circumference <90 cm, patients with waist circumference ≧90 cm were associated with a higher risk of the severe degree of GERD (AOR: 1.264, 95% CI: 1.014–1.582).

In terms of lifestyle habits, patients with an *Areca catechu* chewing habit were positively associated with the mild degree of GERD (AOR: 1.462, 95% CI: 1.121–1.908). Moreover, compared with the patients without sleep disorders, patients with such symptoms were positively associated with the mild degree of GERD (AOR: 1.129, 95% CI: 0.526–0.950), but there was no significant association for drinking, eating and smoking habits.

In addition, in terms of perceived symptoms, patients with otolaryngology symptoms (e.g., tinnitus) were positively associated with the mild degree of GERD compared with the patients without such symptoms (AOR: 1.232, 95% CI: 0.559–0.957). Similarly, while adjusting with all variables, the patients with hepatobiliary and gastrointestinal symptoms (e.g., diarrhea) were associated with a lower risk of the mild degree of GERD (AOR: 0.607, 95% CI: 0.438–0.841), and the patients with such symptoms (e.g., stomachache) were associated with a higher risk of the severe degree of GERD (AOR: 1.846, 95% CI: 1.224–3.329).

## 4. Discussion

This research collected cases of GERD from medical records containing health questionnaires and the report of endoscopic findings on the day of health check-ups. Then, we analyzed the influencing factors of GERD by using a health information questionnaire and the health check-up parameters. In this study, age was a risk factor, and we found that the prevalence of mild degree of GERD was most likely in the middle age group (31–60 years), but decreased in the groups younger than 30 and older than 60 years. These findings were consistent with a previous study [11], suggesting that the prevalence of GERD increases with age in the general population, while decreasing in elderly patients. At present, the reasons for the parabolic distribution of the prevalence of GERD with age are still unclear. However, high psychosocial stress, unhealthy food, lack of exercise and family history may contribute to the highest prevalence of GERD in middle-aged patients.

Sex was found to be a risk factor. In previous studies, Nasseri-Moghaddam et al. [12] examined the epidemiology of GERD and found that sex is a risk factor. Females were 1.55 times more likely to have GERD than males. This finding has been confirmed recently by a meta-analysis that reported the prevalence of GERD by sex was reported in 50 studies, and found that the pooled prevalence of GERD in females was moderately higher than in males [13]. Furthermore, the past studies showed the symptomatic GERD was mainly in females [14], and the difference was more pronounced during peri-menopause [15]. This is consistent with our findings. Although the subjects of these studies are different from the present study, it can be confirmed that sex is indeed a risk factor, especially for females. This study also showed that waist circumference is a risk factor of GERD. The past research has shown that waist circumference is the most important factor for GERD, and females are more likely to suffer from disease [16]. Then, as our study showed, waist circumference increased due to accumulation of fat. This is consistent with another study that have found a higher average visceral fat area is associated with erosive esophagitis, which is caused by external pressure from accumulation of fat around the stomach [17].

In addition, poor lifestyle habits may increase the burden on the stomach, which leads to gastroesophageal reflux symptoms. Our study shows that chewing *Areca catechu* is indeed a risk factor of GERD. The ingredients of areca nut include different alkaloids, such as including guvacine, arecoline, arecoline and guvacoline, which can induce systemic effects, and arecoline can cause cancer [18]. The incidence of GERD was slightly higher in the subjects who chewed *Areca catechu* than in those who did not chew *Areca catechu*. Hsieh et al. [19] showed that the probability of GERD caused by chewing *Areca catechu* daily was 1.496 times higher than that not caused by chewing *Areca catechu* daily (95% CI: 1.02–2.21). Wang et al. [20] showed that people who chewed *Areca catechu* every day were two times more likely to have GERD than those who did not chew *Areca catechu* daily (95% CI: 1.2–3.2). Another previous study found that *Areca catechu* chewing increased the release of histamine and gastric acid, but an abnormal excess of gastric acid is one of the etiologies of reflux esophagitis and peptic ulcer disease [21]. In contrast, a previous study found that smoking, alcohol consumption and *Areca catechu* chewing were independently associated with reflux esophagitis [22]. Another study has pointed out that patients eating fatty, fried, sour, or spicy foods may experience more severe GERD. What’s more, it’s not just what you eat, but how often you eat that can play a role in causing one’s symptoms. Eating too much or too little can be a burden on the stomach [23]. These were different from the results of our study that we only found *Areca catechu* chewing increased the risk of GERD. One possible explanation for the findings is that patients who have regular health checks may be more aware of different symptoms, so are more likely to seek treatment to improve bad habits in mild stages and pay more attention to their eating habits. Another explanation could be that Taiwan’s population has low rates of smoking and alcohol consumption. Besides, people with sleep disorders may suffer from poor sleep quality, which indirectly affects not only the mental system, but also the digestive system. The past study proposed positive associations between GERD, sleep disturbances, and psychological stress [24]. Thus, GERD was a risk factor for sleep disorders, and many studies have assumed that the association seems to be bidirectional [25,26,27,28]. GERD awakens patients during the night, which adversely affects the quality of sleep. On the other hand, sleep disorder may adversely affect GERD by increasing intraesophageal stimulation (e.g., gastric acid) and increasing esophageal exposure to gastric acid, possibly by changing the ghrelin and leptin ratio [28]. Moreover, Carabotti et al. [29] suggested that stress and lifestyle are major risk factors for gastritis, so improving sleep disorders can reduce the incidence of gastritis, and thus indirectly reduce GERD.

Some studies have suggested that sensorineural tinnitus is associated with GERD [30,31]. The prevalence of tinnitus in adults is about 30–40%, and it is related to the psychological stress [32,33]. The present study also found that patients with tinnitus will be more likely to develop GERD. Echoing the above, high stress can lead to many physical problems, including mental illness, which may lead to the risk of GERD. Studies in Western countries have reported that GERD is a common digestive tract disease. In the United States, 10–15% of adults experience heartburn once a day, and 24–33% have at least one episode of heartburn or acid reflux per week. Among the patients with GERD, 14% had no obvious symptoms [34,35]. In other words, although diarrhea may be a symptom of GERD, most people may not notice it because the symptoms are not apparent or think diarrhea may be normal. Furthermore, it may be because patients actually had both GERD and some hepatobiliary and gastrointestinal symptoms, which resulted in a decline in awareness. In contrast, stomachache is a typical symptom of GERD. There was a previous study examined the presence of upper gastrointestinal and extrasoesophageal symptoms. They found that patients with GERD tended to have more frequent and more severe stomachache [36]. The GERD symptoms may be accompanied by a fear of developing stomachaches every day. This may be related to reduced exercise, which is associated with weight gain. Most people who gain weight are more likely to increase their waist circumference. These are related to physical factors that increase the risk of developing GERD [37].

There were some limitations in our study. At first, this study was a retrospective cross-sectional study, and a comprehensive health questionnaire was adopted, so no special health risk questionnaire of GERD was used. Moreover, because anti-reflux drugs are widely used, compliance inevitably varies significantly among different populations. Therefore, it is difficult to determine the actually use of anti-reflux drugs prior to gastroscopy. Since some GERD symptoms (heartburn and regurgitation) were not included in the health questionnaire in this study, we could not distinguish the non-erosive reflux disease (NERD) patient from normal subjects. Hence, these important aspects need further study in the future. Besides, this study was conducted in the health management center of a medical center in southern Taiwan. Owing to the regional and individual differences, the results may not be extrapolated to patients with GERD across the country. In this study, only the patients diagnosed as having GERD by physicians using gastroscopy were evaluated. However, some patients had GERD but did not necessarily have endoscopic manifestations, which might have led to selection bias. The information on the duration of symptoms and treatment before gastroscopy as well as between gastroscopy and patient recruitment is important as they may have a significant effect on esophagus histology and thereby on classification of the disease severity. However, because of the limited information of the medical records (including questionnaires) in this study, we cannot trace back the duration of symptoms nor the treatment if physicians did not record that data. In addition, we used secondary data to examine the correlation and difference of realization, so we could not explain the causal relationship between the independent and dependent variables. Because the original questionnaire is a comprehensive health questionnaire, it is not a special questionnaire to evaluate GERD, therefore, the severity of GERD symptoms cannot be accurately measured. For patients with GERD, anti-reflux medication would be given according to not only the endoscopic severity but also the patient’s symptoms, disease status and lifestyle habits. However, the subjects of this study were patients undergoing routine health check-ups, not outpatients. If there are any problems after the health check-up, they will be referred for outpatient treatment, so it is not possible to track the medication treatment and the outcome, which is a limitation of this study.

## 5. Conclusions

The results of our study show the status and related influencing factors of GERD, which can be used as a reference for public health care and clinicians. Because most GERD cases can be controlled and prevented by lifestyle changes it is important to raise public awareness regarding lifestyle trends that predispose to GERD. This will hopefully lead the public to adopt a healthier lifestyle thus reducing the overall medical burden of GERD. Therefore, the present study provides insights for reducing GERD and suggests that the medical institutions should strengthen the publicity and education of health policies to reduce the prevalence of GERD and thus establish a healthy living environment.

## Figures and Tables

**Table 1 ijerph-18-03808-t001:** Statistics of demographic variables (*n* = 5653).

Demographic Variables	Cases	Percentage
Total	5653	100.0
Normal	2249	39.8
Mild degree	3003	53.1
Severe degree	401	7.1
Sex		
Male	3728	65.9
Female	1925	34.1
Age		
≦30	193	3.4
31–60	4049	71.6
>60	1411	25.0
BMI (kg/m^2^)		
<18.5	218	3.9
≧18.5~<24	2533	44.8
≧24~<35	2865	50.7
≥35	37	0.6
Waist circumference (cm)		
<90	3799	72.4
≧90	1456	27.6

Note: BMI = body mass index.

**Table 2 ijerph-18-03808-t002:** Multivariate logistic regression analysis results (*n* = 5653).

		Mild Degree (Reference: Normal)				Severe Degree (Reference: Normal)			
		Adjusted Odds Ratio	95% Confidence Interval		*p* Value	Adjusted Odds Ratio	95% Confidence Interval		*p* Value
			Lower	Upper			Lower	Upper	
Age	≦30 (Reference)								
	31–60	1.132	0.540	0.993	0.045 *	0.767	0.418	1.408	0.393
	>60	0.808	0.587	1.114	0.194	0.726	0.385	1.370	0.323
Sex	Female (Reference)								
	Male	0.842	0.732	0.969	0.017 *	1.016	0.772	1.336	0.911
BMI	<18.5 (Reference)								
	≧18.5~<24	0.953	0.708	1.283	0.753	1.153	0.679	1.960	0.598
	≧24~<35	0.834	0.612	1.136	0.249	1.383	0.788	2.430	0.259
	≥35	0.629	0.295	1.340	0.229	3.708	0.446	30.839	0.225
Waist circumference (cm)	<90 (Reference)								
	≧90	0.996	0.887	1.117	0.941	1.264	1.014	1.582	0.038 *
Drinking habit	Never (Reference)								
	Occasionally	0.970	0.860	1.096	0.628	1.035	0.818	1.310	0.774
	every day	0.998	0.795	1.253	0.988	0.993	0.642	1.536	0.976
Eating habit	No distinction (Reference)								
	non-vegetarian	0.887	0.752	1.046	0.153	0.946	0.687	1.302	0.733
	vegetarian	0.816	0.565	1.178	0.277	0.760	0.387	1.492	0.425
Smoking habit	No (Reference)								
	Yes	0.946	0.828	1.080	0.409	0.905	0.698	1.174	0.452
Areca catechu chewing habit	No (Reference)								
	Yes	1.462	1.121	1.908	0.005 **	0.965	0.599	1.552	0.882
Sleep disorders	No (Reference)								
	Yes	1.129	0.526	0.950	0.042 *	3.100	0.388	24.783	0.286
Otolaryngology symptoms	Never (Reference)								
	Dizziness	0.970	0.774	1.217	0.794	0.933	0.609	1.431	0.752
	Headache	0.980	0.801	1.198	0.842	0.815	0.563	1.180	0.279
	Tinnitus	1.232	0.559	0.957	0.023 *	0.985	0.567	1.711	0.957
Cardiovascular symptoms	Never (Reference)								
	Chest pain	1.140	0.973	1.335	0.104	1.130	0.829	1.541	0.439
	Palpitation	0.897	0.659	1.221	0.491	0.765	0.444	1.319	0.335
	Fainted before	1.082	0.766	1.528	0.656	0.933	0.494	1.762	0.831
Respiratory symptoms	Never (Reference)								
	Cough	0.859	0.725	1.018	0.080	0.971	0.697	1.352	0.862
	Dyspnea	0.651	0.310	1.366	0.256	0.953	0.208	4.364	0.951
	Asthma	1.166	0.960	1.415	0.121	1.001	0.695	1.440	0.998
Renal urinary symptoms	Never (Reference)								
	Painful urination	1.227	0.785	1.920	0.369	1.190	0.492	2.879	0.700
	Frequent urination	1.020	0.812	1.280	0.868	1.078	0.695	1.673	0.738
	Nocturnal enuresis	0.961	0.787	1.173	0.694	1.228	0.814	1.853	0.327
Hepatobiliary and gastrointestinal symptoms	Never (Reference)								
	Disgusting	0.703	0.430	1.150	0.161	0.571	0.254	1.287	0.177
	Stomach ache	1.038	0.815	1.321	0.762	1.846	1.224	3.329	0.042 *
	Bloating	0.962	0.747	1.239	0.762	0.863	0.545	1.366	0.529
	Constipation	0.939	0.735	1.200	0.617	0.820	0.525	1.280	0.383
	Diarrhea	0.607	0.438	0.841	0.003 **	0.614	0.345	1.093	0.097

Note: BMI = body mass index. * *p* < 0.05, ** *p* < 0.01.

## Data Availability

The data presented in this study are available on request from the corresponding author. The data are not publicly available due to privacy restrictions.

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
