# Peer review of "Associations between Lifestyle Habits, Perceived Symptoms and Gastroesophageal Reflux Disease in Patients Seeking Health Check-Ups"

_ijerph, 2021, doi:10.3390/ijerph18073808_

Round 1

Reviewer 1 Report

The revised manuscript  ijerph-1148174 is acceptable in its current form. However, certain sentences are incomplete or grammatically incorrect. My suggestions for these sentences are as follows:

Lines 30-31. “About 60.2% (n=3404) of patients with GERD by the reports of endoscopic findings.” Suggested change: “About 60.2% (n=3404) of patients with GERD were diagnosed based on endoscopic findings.”

Lines 36-37. “Because most GERD can be controlled and prevented by lifestyle modification, it deserves health professionals pay more attention to individual’s symptoms and lifestyle habits.” Suggested change: “Because most GERD cases can be controlled and prevented by lifestyle modifications, health professionals should always obtain a detailed history regarding symptoms and lifestyle habits associated with GERD”

Lines 251-2. “Weight gain, which in turn increases waist circumference.”

 Incomplete sentence. Please, rephrase.

Lines 263-4. “Moreover, because anti-reflux drugs are widely used, and there will be compliance among different populations. Therefore, the usage of anti-reflux drugs pre gastroscopy is difficult to be traced.” Suggested change: “Moreover, because anti-reflux drugs are widely used, compliance inevitably varies significantly among different populations. Therefore, it is difficult to determine the actually use of anti-reflux drugs prior to gastroscopy”

Line 275. “As the information on the duration of symptoms and treatment ….”. Please delete the “As”

Lines 293-5. “Most GERD can be controlled and prevented by lifestyle changes, so if the public does not face to the occurrence of GERD, it may result in a personal health and medical burden.” Suggested change: “Because most GERD cases can be controlled and prevented by lifestyle changes it is important to raise public awareness regarding lifestyle trends that predispose to GERD. This will hopefully lead the public to adopt a healthier lifestyle thus reducing the overall medical burden of GERD”.

Author Response

Response to Reviewers

We are grateful for the thoughtful comments of reviewers. Your suggestions have enriched the manuscript, and our revision can be clearer and more compelling. These responses to reviewers are shown in the following tables:

Reviewer 1:

Reviewer’s Comments

Authors’ Responses

1

Lines 30-31. “About 60.2% (n=3404) of patients with GERD by the reports of endoscopic findings.” Suggested change: “About 60.2% (n=3404) of patients with GERD were diagnosed based on endoscopic findings.”

Thank you for your suggestions. We have modified this sentence according to your suggestions: “About 60.2% (n=3404) of patients with GERD were diagnosed based on endoscopic findings”. (Lines 30-31.)

2

Lines 36-37. “Because most GERD can be controlled and prevented by lifestyle modification, it deserves health professionals pay more attention to individual’s symptoms and lifestyle habits.” Suggested change: “Because most GERD cases can be controlled and prevented by lifestyle modifications, health professionals should always obtain a detailed history regarding symptoms and lifestyle habits associated with GERD”

We have modified this sentence according to your suggestions: “Because most GERD cases can be controlled and prevented by lifestyle modifications, health professionals should always obtain a detailed history regarding symptoms and lifestyle habits associated with GERD”. (Lines 36-38.)

3

Lines 251-2. “Weight gain, which in turn increases waist circumference.”

 Incomplete sentence. Please, rephrase.

We have rephrased this paragraph according to your suggestions: “Most people who gain weight are more likely to increase their waist circumference”.  (Lines 256-257.)

4

Lines 263-4. “Moreover, because anti-reflux drugs are widely used, and there will be compliance among different populations. Therefore, the usage of anti-reflux drugs pre gastroscopy is difficult to be traced.” Suggested change: “Moreover, because anti-reflux drugs are widely used, compliance inevitably varies significantly among different populations. Therefore, it is difficult to determine the actually use of anti-reflux drugs prior to gastroscopy”

We have modified this sentence according to your suggestions: “Moreover, because anti-reflux drugs are widely used, compliance inevitably varies significantly among different populations. Therefore, it is difficult to determine the actually use of anti-reflux drugs prior to gastroscopy”. (Lines 261-264.)

5

Line 275. “As the information on the duration of symptoms and treatment ….”. Please delete the “As”

We have deleted the “As” according to your suggestions. (Lines 272-275.)

6

Lines 293-5. “Most GERD can be controlled and prevented by lifestyle changes, so if the public does not face to the occurrence of GERD, it may result in a personal health and medical burden.” Suggested change: “Because most GERD cases can be controlled and prevented by lifestyle changes it is important to raise public awareness regarding lifestyle trends that predispose to GERD. This will hopefully lead the public to adopt a healthier lifestyle thus reducing the overall medical burden of GERD”.

We have modified this sentence according to your suggestions: “Because most GERD cases can be controlled and prevented by lifestyle changes it is important to raise public awareness regarding lifestyle trends that predispose to GERD. This will hopefully lead the public to adopt a healthier lifestyle thus reducing the overall medical burden of GERD”. (Lines 290-293.)

Reviewer 2 Report

Dr. Chang and colleagues conducted a retrospective study to investigate the association of lifestyle habits, perceived symptoms and GER in patients who underwent routine health check-up.

Although the study population was large and the issue is interesting, in my opinion there are methodological flaws that limit the significance of the results.

Firstly, I cannot understand the reason why an EGDS was performed in routine health checkup (line 83). Was it really done in all patients irrespectively of symptoms? The justification for such an invasive screening should be clearly defined. Otherwise, if EGDS was performed in a selected subgroup of patients this should be outlined.

Were any other investigations performed to better define GER? The authors just analyzed patients with endoscopic inflammation according to Los Angeles criteria, but non-erosive reflux disease (NERD) and other entities that could be associated with GER-like symptoms were not considered. Indeed, NERD patients represent up to 60% of all patients with reflux symptoms.

Moreover, only 10% of questionnaires were verified, therefore we cannot be confident about the quality of data.

Author Response

Response to Reviewers

We are grateful for the thoughtful comments of reviewers. Your suggestions have enriched the manuscript, and our revision can be clearer and more compelling. These responses to reviewers are shown in the following tables:

Reviewer 2:

Reviewer’s Comments

Authors’ Responses

1

Firstly, I cannot understand the reason why an EGDS was performed in routine health checkup (line 83). Was it really done in all patients irrespectively of symptoms? The justification for such an invasive screening should be clearly defined. Otherwise, if EGDS was performed in a selected subgroup of patients this should be outlined.

Thank you for your comments. We did not perform EGDS for all health check-up participants. Only those who received EGDS with complete medical records were included in this study. We revised the section of the Method as follows: “This study only collected medical records from health check-up participants who have received gastroscopies with complete medical records”. (Lines 83-84.)

2

Were any other investigations performed to better define GER? The authors just analyzed patients with endoscopic inflammation according to Los Angeles criteria, but non-erosive reflux disease (NERD) and other entities that could be associated with GER-like symptoms were not considered. Indeed, NERD patients represent up to 60% of all patients with reflux symptoms.

We thank for the reviewer's wise suggestion. Considering the limitation of our questionnaire that didn't include GERD symptoms, we also used the NBI endoscopy in improving the GERD diagnostic accuracy.  We revised the section of “Method” as follows: “In health management center, all gastroscopy exam were performed with narrow band image (NBI) to improve GERD diagnosis accuracy”. (Lines 97-98.)

3

Moreover, only 10% of questionnaires were verified, therefore we cannot be confident about the quality of data.

Thank you for your advice. In general, we conducted the data validation in three stages.

First, each data was checked, and the incidence and severity of GERD found on gastroscopy were recorded in detail.

Second, three gastroenterologists from the gastroenterology department of the medical center were invited to compare and verify the randomly sampled data according to the method suggested by Worster and Haines (2004). After verification, any entries that were found to be written too lightly were immediately compared, corrected, and logged.

Finally, in order to improve the accuracy of the data, three physicians were invited again to jointly select 10% of the samples in the same way, and 565 samples were selected for the second verification. After verification, no data errors were found. The logged data in this study were 100% consistent. (Lines 95-106.)

Reviewer 3 Report

IJERPH-1148174. 16th of March 2021.

This article by Chang et al is exploring gastroesophageal reflux disease in a cross-sectional and retrospective setting. Authors should consider revising their manuscript before publication, as several issues need to be amended. These are outlined below.

Lines 75-107:

I have several concerns about ethical approval. Why was written consent waived ? (lines 106-107 and supplementary material). Besides, the Ethics committee is stated as “Kaohsiung Veterans General Hospital”. Therefore, is the hospital only visited by ex-military personnel ? If so, this would constitute a selection bias.

Another major concern is the endoscopy exploration of the esophagus. How was this performed ? Were patients sedated ? If not, please justify. Authors need to explain in more details the medical procedures performed on patients.

Moreover, how can the authors justify that a health check-up contains endoscopy, which is a rather invasive medical exploration. (lines 82-83). Were underlying conditions suspected ?

Finally, from the supplementary material provided, authors should explicitly state in the methods that health checkups were self-funded by patients.

Line 174, Line 291:

Discussion should be section 4, not section 1. Please edit.
Conclusion should be section 5, not section 1. Please edit.

Lines 256-259:

This sentence should be removed, since it is misleading.

Lines 265-266 :

What is “screw data” ? Besides, it is difficult to understand what the authors mean by “cognitive bias” regarding the use of anti-reflux medications. Authors should clarify and explain the meaning of this sentence.

Lines 21-298:

Please explain what is Areca catechu. Is it a plant extract ? Like tobacco ? Is it derived from animal sources ? Authors need to explicit technical details more thoroughly. Although this is partly explained in the discussion (lines 203-205), it needs to appear earlier in the manuscript, and in more details.

Finally, I suggest revising the English throughout the manuscript.

Author Response

Response to Reviewers

We are grateful for the thoughtful comments of reviewers. Your suggestions have enriched the manuscript, and our revision can be clearer and more compelling. These responses to reviewers are shown in the following tables:

Reviewer 3:

Reviewer’s Comments

Authors’ Responses

1

I have several concerns about ethical approval. Why was written consent waived ? (lines 106-107 and supplementary material). Besides, the Ethics committee is stated as “Kaohsiung Veterans General Hospital”. Therefore, is the hospital only visited by ex-military personnel ? If so, this would constitute a selection bias.

Thank you for your comments. Waived of written consent is applicable to medical record review study, as designed in this study. This type of study is of minimal risk and can’t identify specific personal information obtained from legitimate databases. We have added the explanation to our manuscript. (Lines 110-111.) Besides, Kaohsiung Veterans General Hospital is the largest government medical center in southern Taiwan and is open to all citizens, not limited to ex-military personnel.

2

Another major concern is the endoscopy exploration of the esophagus. How was this performed ? Were patients sedated ? If not, please justify. Authors need to explain in more details the medical procedures performed on patients.

Thank you for your suggestions. All patients were sedated so that gastroscopy could be performed. In addition, if the patient was uncomfortable during the procedure and the medical report will show “incomplete procedure” without any diagnosis.  Therefore, our study did not include incomplete medical records. This study collected medical records from health check-up participants who have received gastroscopies with complete medical records. Their medical records containing the health questionnaire and the report of endoscopic findings were collected for analysis. (Lines 83-85.)

3

Moreover, how can the authors justify that a health check-up contains endoscopy, which is a rather invasive medical exploration. (lines 82-83). Were underlying conditions suspected ?

We thank for the reviewer for pointing out the comments. We did not perform gastroscopy for all health check-up participants. Only those who received gastroscopy with complete medical records were included in this study. We revised the section of the Method as follows: “This study only collected medical records from health check-up participants who have received gastroscopies with complete medical records”. (Lines 83-84.)

4

Finally, from the supplementary material provided, authors should explicitly state in the methods that health checkups were self-funded by patients.

Thank you for your suggestions. We have modified this sentence according to your suggestions: “The health check-up were self-funded by patients”. (Lines 79-80.)

5

Line 174, Line 291:

Discussion should be section 4, not section 1. Please edit.
Conclusion should be section 5, not section 1. Please edit.

Thank you for reminding us. After our examination, we found that our original manuscript was correct. Maybe the journal assistant didn't notice during the typesetting process. We have modified the section. (Line 179, Line 288.)

6

Lines 256-259:

This sentence should be removed, since it is misleading.

Thank you for your advice. We have deleted the part you mentioned. (Lines 259-262.)

7

Lines 265-266 :

What is “screw data” ? Besides, it is difficult to understand what the authors mean by “cognitive bias” regarding the use of anti-reflux medications. Authors should clarify and explain the meaning of this sentence.

We apologize for the confusion. We deleted the sentence that you mentioned to reduce the chance of misleading. (Lines 261-266.)

8

Lines 21-298:

Please explain what is Areca catechu. Is it a plant extract ? Like tobacco ? Is it derived from animal sources ? Authors need to explicit technical details more thoroughly. Although this is partly explained in the discussion (lines 203-205), it needs to appear earlier in the manuscript, and in more details.

Thank you for your suggestions. Areca catechu is actually “betel nut”. We originally used the term "betel nuts chewing habit". In the last revision, a reviewer suggested that we refer to it by Latin name. According to your suggestions, we added a note when "Areca catechu" first appears. (Lines 123-124.)

9

Finally, I suggest revising the English throughout the manuscript.

Thank you for your advice. We already submitted our manuscript for English editing. Please see the Appendix for the certificate.

Round 2

Reviewer 2 Report

Although the authors have tried to improve the manuscript, in my opinion the methodological flaws already highlighted cannot guarantee the publication in this journal.

Since the criteria for undergoing EGDS are not specified, the prevalence of GER-like symptoms is unclear, the reliability of data is unknown, the lack of analysis on other forms of GER other than the classical, erosive one, my recommendations remains unchanged.

I am sorry for this, and think that probably the authors will be able to focus their clinical question a bit more and to find an appropriate journal.

Author Response

Response to Reviewers

We are grateful for the thoughtful comments of reviewers. Your suggestions have enriched the manuscript, and our revision can be clearer and more compelling. These responses to reviewers are shown in the following tables:

Reviewer 2:

Reviewer’s Comments

Authors’ Responses

Although the authors have tried to improve the manuscript, in my opinion the methodological flaws already highlighted cannot guarantee the publication in this journal.

Since the criteria for undergoing EGDS are not specified, the prevalence of GER-like symptoms is unclear, the reliability of data is unknown, the lack of analysis on other forms of GER other than the classical, erosive one, my recommendations remains unchanged.

I am sorry for this, and think that probably the authors will be able to focus their clinical question a bit more and to find an appropriate journal.

Thanks for the comments.

In this study, for all of the enrolled patients, the purpose of EGDS examination were only for health check-up rather than medical need. The criteria for undergoing EGDS were peoples without contraindication for receiving analgesia for painless endoscopy in this study. The methodology in this study was a retrospective study by using chart review on the data of medical record and questionnaires. The questionnaires designed to survey general medical conditions for general population and filled by peoples for the present health examination not for study design.

In addition, according to our previous publication by using the similar data as this study, we provided solid and valuable information for the prevalence of Barrett’s esophagus (Chen et al., 2019). In this study, focusing on the other aspect of GERD, the prevalence of GERD-like symptoms and the reliability of our data should be without doubt.

On the other hand, in this study, we focused on the impact of life style habits and perceived symptoms on the patients with GERD. Because of the limitation of the objective definition for the non-erosive reflux esophagitis by retrospective review, we just enrolled the patients who with positive endoscopic findings as GERD for the prevention of the selection bias.

Therefore, we wish our statement could dispel all the misunderstandings for you.

In addition, we also request your last reviewing again for this study results. Hope you can give us the positive response to our study.  Thank you.

Reference:

Chen, Y. H.; Yu, H. C.; Lin, K. H.; Lin, H. S.; Hsu, P. I. Prevalence and risk factors for Barrett's esophagus in Taiwan. World journal of gastroenterology 2019, 25, 3231–3241.

Reviewer 3 Report

The authors have answered all of my concerns.
Please edit the English, again, as some mistakes are still present. MDPI will help you in doing so.

Author Response

Response to Reviewers

We are grateful for the thoughtful comments of reviewers. Your suggestions have enriched the manuscript, and our revision can be clearer and more compelling. These responses to reviewers are shown in the following tables:

Reviewer 3:

Reviewer’s Comments

Authors’ Responses

The authors have answered all of my concerns.

Please edit the English, again, as some mistakes are still present. MDPI will help you in doing so.

Thank you for your recommendation. We have already provided the certificate of Editage English editing for our manuscript (please see the appendix).

According to assistant editor Ms. Stealu Shi’s comment: “after the article is accepted, we will provide English revision services”. We will submit MDPI for English editing after our manuscript is accepted for publication. Hope you can give us this opportunity to enter the English editing program.

This manuscript is a resubmission of an earlier submission. The following is a list of the peer review reports and author responses from that submission.

Round 1

Reviewer 1 Report

Dear Editor

Thank you for giving me this paper to review. 
I must point, this article in its present form is not acceptable to publish in IJERPH.

The strengths - All patients had an endoscopy. It makes population homogeny. - Form of paper – well divided The weakness - Title. It’s too long and doesn’t tell about this study. - Lack of information regarding pharmacotherapy after endoscopy. - Unappropriated described study design. - The results part is too short and must be well explained. The alone table is insufficient. - There aren’t well-described degrees of GERD (low, mild, and severe) - As I mentioned before, betel nuts chewing habit. This habit must be better described. Dear Authors, please define the Latin name of this plan and explain why it is so dangerous. - Generally, lifestyle habits should be described in the discussion. - This work must be modified and well written.

I am waiting for a modified version of this manuscript. 

Sincerely 

Reviewer 2 Report

The authors conducted a retrospective study to evaluate the factors associated with GERD diagnosed by gastroscopy. The strength of this study is the fact that all patients had undergone endoscopy. However, due to the retrospective design, there are certain drawbacks, including the lack of important information on medications used before and after gastroscopy, details concerning the life style parameters, which limit the value of the findings. Title and English need revision.

Comments

  1. Title
  • The term “effect” implies causality which cannot be assessed in a retrospective study design. “association” is a more suitable term. This is also the case for the other parts of the text where the terms “effect” and “influence” are used.
  • Apparently, the authors using the term “health status” mean the life style. Since the term health status usually refers to the disease status, relevant changes are suggested in the title and throughout the manuscript.

  1. Introduction

Lines 76-8. “The specific objective of this study was to examine the “effect” of perceived health status, symptoms, and diseases on GERD in patients who underwent physical examination”. Please change to: “The specific objective of this study was to examine the association of perceived health status, symptoms, and diseases with GERD in patients who visited the outpatient clinics”.

  1. Methods
  • The study design should be clearly defined
  • The description of “Data sources and study samples” is not clear. The following issues need further clarification:
  • Which part of the study was retrospective and which (if any) was prospective?
  • At which time point the questionnaire was distributed and how it was completed (by the patient, in the presence of physician-researcher??) ?
  • How GERD was defined clinically? This is important since endoscopy findings were normal in some of the participants.
  • Perceived health status variables (drinking, eating, smoking, and betel nut chewing habits)”. These variables represent life style rather than health status.

  1. Results
  • For description of the results the past tense should be consistently used.
  • Regression analysis shows associations (significant or not; positive or inverse) rather than differences.

  1. Discussion
  • As gastroscopy-histology findings are central in the study design, information on the duration of symptoms and treatment before gastroscopy as well as between gastroscopy and patient recruitment is important as they may have a significant effect on esophagus histology and thereby on classification of the disease severity. The authors should discuss the potential effect of these missing data on their results.
  • Lines 172-3. “This research collected cases of GERD confirmed with gastroscopy during physical examination”. Do the authors mean that cases were collected during their visit to the out-patient clinics.
  1. Line 198 “Poor lifestyle habits such as chewing betel nuts”. Chewing betel nuts poorly reflects the life style. The authors could discuss the lack of association between GERD and important parameters of life style, such as drinking, eating, smoking, and betel nut chewing habits.
  • Lines 198-206. The paragraph discussing the association with chewing betel nuts (and other parameters of life style) would better move further beyond, so that it will not interfere between the comments regarding diseases and symptoms.
  • Lines 175-6. “Nasseri‐Moghaddam et [11] examined the epidemiology of GERD and found  that  sex  is a risk factor.  Females were  1.55 times  more  likely to have GERD  than males.” This finding has been confirmed recently by a cross-sectional study from Saudi Arabia (Halawani  et al 2020) and a meta-analysis (Nirwan  et al 2020) that reported a moderate predominance of women. Suggested references “Nirwan JS, Hasan SS, Babar ZU, Conway BR, Ghori MU. Global Prevalence and Risk Factors of Gastro-oesophageal Reflux Disease (GORD): Systematic Review with Meta-analysis. Sci Rep. 2020 Apr 2;10(1):5814. doi: 10.1038/s41598-020-62795-1. PMID: 32242117; PMCID: PMC7118109.” “Halawani H, Banoon S. Prevalence and Determinants of Gastroesophageal Reflux Disease and the Risk Factors Among Adult Patients Attending Al-Iskan Primary Health Care Center in Makkah, 2020. Cureus. 2020 Sep 18;12(9):e10535. doi: 10.7759/cureus.10535. PMID: 33094075; PMCID: PMC7574976”.
  • The negative association between mild GERD and sleep disturbances found in this study is surprising. In lines 223-4, the authors comment “However, there is little evidence of a causal relationship between sleep disorders and GERD in the existing literature and these have shown inconsistent results”. This comment contrasts the bulk of literature showing a positive association between GERD and sleep disturbances and describe the pathophysiological mechanisms underlying this association. Do the authors have a potential explanation for their finding?? (A recent relevant article: Orr WC, Fass R, Sundaram SS, Scheimann AO. The effect of sleep on gastrointestinal functioning in common digestive diseases. Lancet Gastroenterol Hepatol. 2020 Jun;5(6):616-624. doi: 10.1016/S2468-1253(19)30412-1. PMID: 32416862)
  • Lines 251-2. “Owing to the measure to preserve the data integrity of the database, we could not fully analyze and evaluate the severity of GERD”. My question is: how fully analysis and evaluation of the severity of GERD would affect data integrity of the database ??
  • Lines 256-7. Conclusions: “Especially for the controllable factors of GERD, such as betel nut chewing habit, the incidence of GERD can be reduced as long as the general population changes their health-related behaviors and adjust their daily lifestyles”. Although it is generally accepted that the life style is an important risk factor of GERD, only the association with betel nut chewing habit cannot support this part of the conclusions.

Additional comments

  • Line 83-4. “After physical examination and gastroscopy, all of the patients received doctor’s intervention and making conclusion for  not only  endoscopic findings but also other results such as laboratory or images at the end of health examination in the same day”. Please rephrase
  • Line 208. “Tinnitus has  a  high  risk of  30-40%  in  adults …”. Please change “high risk” with “prevalence”
  • Line 219. “which resulted in a decline in consciousness”. Suggested change “which resulted in a decline in awareness”.
  • Lines 229-31. “At first, because of the retrospective in nature of this study as well  as  the health risk questionnaire of GERD that been not performed in our routine practice in health examination, we did not provide this information in this study”. Please rephrase this sentence and clarify “this information”.
  • Line 240. “the usage of anti-reflux  medication pre-endoscopy is hard to be traced because of the wide spectrum of drugs population for antireflux medication”. Please rephrase.

Reviewer 3 Report

The manuscript by Vhang et al investigated the risk factor associated with GERD. The point of view and the finding that the relationship between betel nut chewing habit and GERD are interesting. However, there are some constitutional problems in this paper.

  1. The title of this manuscript is inadequate. I understand the importance of gastroscopy to diagnose the degree of reflux esophagitis. However, this point is not informed to readers in this title.

  1. The term of hepatobiliary and gastrointestinal diseases is ambiguous. Many diseases of both benign and malignant are included in hepatobiliary and gastrointestinal diseases. More detailed classification of diseases should be performed.

  1. Some GERD patients have no symptoms, others have reflux symptoms such as heartburn without mucosal lesions endoscopically. These non-erosive reflux disease (NERD) patents should be distinguished from normal subjects. This point should be described in discussion.